# State of Mental Health Research of Adolescents and Youth in Chile: An Ontological Analysis

**DOI:** 10.3390/ijerph19169889

**Published:** 2022-08-11

**Authors:** Vania Martínez, Marcelo A. Crockett, Ajay Chandra, Sarah Shabbir Suwasrawala, Arkalgud Ramaprasad, Alicia Núñez, Marcelo Gómez-Rojas

**Affiliations:** 1Centro de Medicina Reproductiva y Desarrollo Integral del Adolescente (CEMERA), Facultad de Medicina, Universidad de Chile, Santiago 8380455, Chile; 2Millennium Nucleus to Improve the Mental Health of Adolescents and Youths (Imhay), Santiago 8380455, Chile; 3Millennium Institute for Research in Depression and Personality (MIDAP), Santiago 7820436, Chile; 4Escuela de Salud Pública, Universidad de Chile, Santiago 8380453, Chile; 5Ramaiah Public Policy Center, MS Ramaiah University of Applied Sciences, Bengaluru 560054, India; 6Information and Decision Sciences Department, University of Illinois at Chicago, Chicago, IL 60607, USA; 7Department of Management Control and Information Systems, School of Economics and Business, Universidad de Chile, Santiago 8330015, Chile

**Keywords:** mental health, research, adolescent, youth, ontology, Chile

## Abstract

Knowing the state of mental health research in adolescents and youth can be an important tool for decision-making, especially in contexts of limited resources. The aim of this study is to map the scientific research on adolescent and youth mental health in Chile using an ontological framework. We have mapped the population of research articles on mental health of adolescents and youth in Chile in Scopus, Web of Science, and SciELO databases onto the ontology. The PRISMA reporting guidelines were used to screen the 1688 items based on relevance, duplication, and version. The corpus of 346 articles was coded into the ontology through an iterative process among the seven authors. This ontological mapping shows isolated research efforts that have been carried out in Chile to explain the whole state of mental health in adolescents and youth. There is a lack of coordination between the priorities established by the decision-makers and the researchers. Our results coincide with the need to strengthen mental health research in the country, and to prioritizing those topics that contribute to decision-making based on the needs of the population.

## 1. Introduction

Mental health problems during adolescence and youth are highly prevalent globally. It is estimated that approximately 20–25% of young people will have mental health disorders each year [1]. In fact, most mental health problems are known to begin during adolescence [2,3,4], with an estimated 50% of them beginning before 14 years of age, and 75% before 24 years of age [2]. At the same time, mental health problems during youth have higher persistence [3]: almost 60% of individuals who present symptoms during adolescence will also present symptoms during early adulthood [5]. Thus, adolescence is considered a high-risk stage for the onset of mental health problems, which can have a negative long-term impact on physical health and personal well-being, at work, and in the educational environment [6,7,8]. In addition, mental health problems during adolescence and youth are one of the main causes of disability-adjusted life years in the world [9], and are considered a public health challenge [1].

Mental health problems in young people are a major concern in Chile due to their high burden of disease [10]. In adolescents, an epidemiological study reported that 16.5% had one or more psychiatric disorders during the last 12 months [11]. In college students, pre-pandemic estimates showed a sustained increase in mental health symptoms [12]. During the SARS-CoV-2 pandemic, a recent study reported that 37.1% of college students had elevated symptoms of depression, 37.9% of anxiety, and 54.6% of stress [13].

A country’s adolescent and youth mental health care management system is necessarily complex, commensurate with the complexity of the problem it must address. A comprehensive management strategy must specify the stage of intervention, the subject’s condition, the population and age group, and the health care delivery setting. Not all strategies may be equally effective. Research is needed to discover effective strategies to reinforce them, ineffective ones to redirect them, and innovative ones to experiment with them.

There has been an increase in scientific research production regarding mental health [14]. However, it is still considered a major challenge that there is no systemic roadmap for research and systematic translation of knowledge and subsequent implementation of potential solution strategies into public policy [15,16]. In Chile, there is not a clear connection between mental health research and decision-making [17,18]. Consequently, one of the objectives of the National Mental Health Plan 2017–2025 is to strengthen the collaboration between researchers and policy makers for the design, implementation, and evaluation of public policies in mental health [18].

Knowing the state of mental health research in adolescents and youth can be an important tool for decision-making, especially in contexts of limited resources and a COVID-19 pandemic that continues to be a major public health threat. It can contribute to establishing research priorities, distributing resources, and designing evidence-based interventions for the prevention and promotion of good mental health. The aim of this study is to map the scientific research on adolescent and youth mental health in Chile using an ontological framework. It will help describe the domain and the relationships among its parts. Ontological frameworks are effective in providing a systemic view and can efficiently formalize, standardize, and manage the available information. They are good tools for mapping the existing literature and describing potential gaps in the current approaches (see for example [19,20,21]).

In the following, we first describe an ontology of mental health of adolescents and youth. Second, we map the corpus of research on the topic in Chile onto the ontology. Third, we discuss the emphases and gaps in the corpus. Fourth, and finally, we propose a roadmap for research to address the problem in a systemic and systematic way. The method is similar to that used to study mental health care during and post COVID-19 [22], COVID-19 vaccine strategies [23], public health informatics research [20], healthcare systems [24], mHealth research [19], and children with disabilities [25].

Ontology of Mental Health of Adolescents and Youth

An ontology of mental health of adolescents and youth is presented in Figure 1. Its five dimensions are represented by the five columns. The elements of each dimension are presented as a taxonomy within the column. We describe the dimensions and the corresponding elements below.

The state of mental health of adolescents and youth may be a consequence of many psychopathological conditions, other conditions, and disorders. These are listed in the ‘Condition’ [26] column of the ontology. Youths can be divided into early (10–14 years), middle (15–17 years) and late (18–19 years) adolescents, and young adults (20–24) based on their age [27]. The age groups are listed under the ‘Unit’ column of the ontology—these constitute the population units of research studies. These adolescents may belong to different ‘Populations’ listed under the third column of the ontology. They may be urban, rural, underprivileged, indigenous, disabled, LGBT (lesbian, gay, bisexual and transgender), and immigrants. If we combine the three columns of Condition, Population, and Unit, there are 25 × 7 × 4 = 700 potential descriptors of adolescents and youth mental health. The descriptors include: (a) eating disorders in urban youth, (b) suicide risk in rural middle adolescents, and (c) internalizing disorders in underprivileged late adolescents.

The different stages of research for adolescents and youth with these conditions are listed under the ‘Stage’ column of the ontology [28]. The place, individual/s or method providing the information or performing the intervention are listed in the ‘Setting’ column of the ontology [28].

In sum, the ontology encapsulates 8 × 25 × 7 × 4 × 15 = 84,000 pathways to describe the research about mental health of adolescents and youth. A pathway is a concatenation of an element from each dimension with the connecting words/phrases adjacent to the dimension. Three illustrative pathways are: (a) treating depressive disorders in urban youth by psychologist, (b) identifying suicide risk in underprivileged middle adolescents by school, (c) preventing substance-related disorders in rural late adolescents by parent. Thus, the ontology: (a) provides a unified framework to address the problem, (b) defines the elements, dimensions, and boundaries of the problem, and (c) encapsulates the combinatorial complexity of the problem. Therefore, it can help address the problem systemically and systematically.

## 2. Materials and Methods

We mapped the population of research articles on mental health of adolescents and youth in Chile onto the ontology. We used a TITLE-ABSTRACT-KEYWORDS search in Scopus and Web of Science databases to obtain the corpus of research. We used a similar search strategy (syntax) to obtain the research corpus from the SciELO (Scientific Electronic Library Online) database. SciELO database covers research articles from Latin America (Argentina, Brazil, Chile, Columbia, Costa Rica, Cuba, Mexico), Spain, Portugal, the Caribbean and South Africa. Since SciELO does not have the TITLE-ABSTRACT-KEYWORDS filter option, the option of ‘all fields’ was selected and articles relating to Chile were filtered and included in the corpus.

We experimented with different search terms before finalizing the following. In each iteration, we studied the items that were included/excluded and based on that feedback modified the search strategy. The final search strategy “TITLE-ABS-KEY ((((“mental health” OR “mental disorder” OR “mental illness” OR “Psychopathology” OR “Neurodevelopmental” OR “Schizophrenia” OR “Bipolar” OR “depression” OR “Autism” OR “Anxiety” OR “OCD” OR “Trauma” OR “Dissociative Disorder” OR “Somatic Symptom” OR “Eating Disorder” OR “Sleep-wake disorder” OR “Sleep disorder” OR “Sexual Dysfunction” OR “Gender Dysphoria” OR “Conduct disorder” OR “Substance-related” OR “drug abuse” OR “Neurocognitive disorder” OR “Personality” OR “Suicide” OR “Stigma” OR “Anorexia” OR “Bulimia” OR “ADHD” OR “Psychotic” OR “Elimination disorder” OR “Paraphilic disorder” OR “Quality of life” OR “Wellbeing”) AND (adolescent OR youth OR “young people” OR student))) AND Chile*)) retrieved a total of 1688 items on 5 August 2021 (SciELO: 241; Scopus: 948; and Web of Science: 499 items).

We used PRISMA reporting guidelines to screen the 1688 items based on relevance, duplication, and version. The details of the same are shown in the PRISMA flowchart (Figure 2). The reference management software Zotero (Corporation for Digital Scholarship, Vienna, VA, USA) was used to screen the corpus from the three databases (automatic screening). During this step, 36 items without abstracts were excluded, 8 duplicate items from within each database were excluded, 270 duplicates items across the three databases were excluded, and 119 items across the databases were merged. Thus, step one of automatic screening yielded 1255 items (i.e., SciELO: 169, Scopus: 613 and Web of Science: 473 items).

The authors downloaded the title, abstract, and keywords of the remaining 1255 items and imported them into an Excel spreadsheet. The corpus was divided into twenty-one approximately equal sets (corresponding to twenty-one pairs amongst the seven coders/authors) and assigned to each pair of coders who coded the articles independently. In a given pair, each of the coders reviewed, discussed, and arrived at a final consensus coding to ensure validity and reliability. After the final coding the articles were further screened manually by four of the authors for repetitions/duplicates, and language variations across the three corpuses. During this step, 54 duplicate items and 855 not relevant items (based on inclusion criteria: age group and country of study) were excluded. Thus, the remaining 346 items were retained and further analyzed.

Each article was coded for the presence/absence of each element of the ontology. The coding was binary (1 for present, 0 for absent) and was not scaled or weighted for multiple occurrences in an article. A glossary of elements was used to ensure the validity of coding. The coding of all the articles went through two iterations by each of the seven authors to ensure its reliability and validity. After the first round of individual coding, the coders discussed the discrepancies in their coding and arrived at a consensus for the final coding. Only the dimensions and elements explicitly articulated in the title, abstract, and keywords were coded. Elements that were implicit in the articles were not coded. In the analysis, both presence and absence of elements convey equally important information.

The coding was used to generate the monad and theme maps to visualize the relative emphasis on the different elements, dimensions, and themes in the corpus as described next. Subsequently in the paper we discuss the implications of the emphases and the gaps in them for addressing the mental health needs of adolescents and youth in Chile.

## 3. Results

We present the monad map (Figure 3) and theme map (Figure 4) below. They are described next.

### 3.1. Monad Map

The monad map in Figure 3 numerically and visually summarizes the frequency of occurrence of each dimension and element of the ontology in the corpus. The number adjacent to the dimension name and the element is the frequency of its occurrence in the 346 journal abstracts that were reviewed and mapped. The bar below each element is proportional to the frequency relative to the maximum frequency among all the elements. Since each item can be coded to multiple elements of a dimension, the sum of the frequency of occurrence of elements may exceed the frequency of occurrence of the dimension to which the elements belong.

The dominant focus is on the dimension of Stage (345), followed by Unit (338), Condition (329), and Setting (248). There is less focus on Population (164). The research articles cover most of the Stage elements with widely ranging emphasis. The dominant focus is on assessing (283) followed by moderate focus on identifying (78). There is far less focus on treating (21). There is least focus on preventing (8), sensitizing (4), diagnosing (4), and counseling (1). There is no emphasis on rehabilitating (0).

There is dominant focus on psychopathology-depressive disorders (133) as a mental health condition. There is moderate emphasis on psychopathology-anxiety disorders (83), psychopathology-substance-related disorders (78), other-suicide risk (49), psychopathology-trauma and stress-related (45), other-wellbeing (38), quality of life (29), psychopathology-impulse control and conduct disorders (27), psychopathology-neurodevelopmental (29), psychopathology-schizophrenia spectrum (23) and psychopathology-eating disorders (23). There is less emphasis on disorder-internalizing (12), disorder-externalizing (11), psychopathology-personality-disorders (10), psychopathology-bipolar and related disorders (8), psychopathology-somatic symptom and related disorders (8), psychopathology-sleep wake disorders (7), other-stigma in mental health (5), psychopathology- obsessive compulsive and related disorders (4), psychopathology-dissociative disorders (1), and psychopathology-neurocognitive disorders (1). There is no emphasis on psychopathology-elimination disorders (0), psychopathology-sexual dysfunctions (0), psychopathology-gender dysphoria (0), psychopathology-paraphilic disorders (0).

There is significant emphasis on different age groups of adolescents. The dominant focus is on late adolescents (221), middle adolescents (215) and early adolescents (202). There is moderate emphasis on young adulthood (149).

There is dominant emphasis on the urban (141) population. There is moderate emphasis on the underprivileged (22) population. There is least emphasis on rural (9), indigenous (8), immigrants (7), LGBT (5), and disabled (2) populations.

Among the care Setting, there is dominant focus on institution-school/university (192). There is moderate focus on institution-clinic (25) and other-parent (18). There is least emphasis on professional doctor (9), psychologist (9), other-family (7), professional mental health worker (4), professional educator (4), other digital/online (2) and professional social worker (1). The research articles do not emphasize other-friend (0), other- volunteer (0), and other-self (0).

### 3.2. Theme Map

The theme map visually summarizes the co-occurrence of elements of the ontology in the population of articles. Hierarchical cluster analysis was done using SPSS (Statistical Package for Social Sciences; IBM: Chicago, IL, USA) with simple matching coefficient (SMC) as the distance measure and the nearest-neighbor aggregation procedure. The detailed rationale for the choice of the clustering method and the presentation of the results are given in La Paz et al. [29] and Syn and Ramaprasad [30]. The four themes represent the four equidistant clusters in the dendrogram of the agglomeration [30]. The colors in Figure 4 highlight the elements of the four themes.

The primary research theme (in red) is assessing of urban early adolescent, middle adolescent, late adolescent, and young adult by institution-school/university. It is a short segment of many potential pathways for mental health of adolescents and youth in the ontology. The potential pathways may include different conditions and populations. The primary research theme is three dimensional and four levelled.

The secondary research theme (in brown) is identifying depressive, anxiety, and substance-related psychopathology disorders, but not with reference particular populations, units, or settings. The potential pathways may include different conditions, unit, and settings. The secondary research theme is two-dimensional but three-levelled.

The tertiary research theme (in yellow) represents psychopathology-trauma and stress-related, psychopathology-impulse control and conduct disorders, other-suicide risk and other-wellbeing. It is a single associated pathway among elements of the Condition dimension. The theme lacks systemic connections with other dimensions. The potential pathways may include different stages, populations, units, and setting for mental health care of adolescents and youth. Thus, the tertiary theme is one-dimensional and four-levelled.

The quaternary research theme (no color) summarizes the little or no research in the research corpus. Sensitizing, diagnosing, preventing, counseling, treating, and rehabilitating are not part of any theme, although assessing and identifying are part of the primary and secondary themes respectively. Similarly, psychopathological conditions like neurodevelopmental, schizophrenia spectrum, bipolar and related disorders, obsessive-compulsive and related disorders, dissociative disorders, somatic symptom and related disorders, eating disorders, elimination disorders, sleep-wake disorders, sexual dysfunctions, gender dysphoria, neurocognitive disorders, personality disorders, paraphilic disorders; other quality of life, and stigma in mental health; and disorder-internalizing and externalizing are all not part of any theme. While psychopathological conditions like—depressive disorders, anxiety disorders and substance-related disorders are part of the secondary theme. Further, the research corpus is focused on urban population as part of the primary theme. However, elements like rural, underprivileged, indigenous, disabled, LGBT and immigrants’ population are not part of any theme. Lastly, the theme highlights the exclusion of a large variety of institutions (clinic, and hospital), professionals (doctor, psychologist, mental health worker, social worker, psychiatric nurse, and educator), others (parent, family, friend, volunteer, and self) and digital/online setting from the research corpus.

Overall, the themes are in decreasing order of dominance in the research—the primary theme is the most dominant and the quaternary theme is absent. The three most dominant themes are one- to four-dimensional, and two- to four-levelled. The quaternary is a vast domain across many dimensions and levels that is yet to be researched.

## 4. Discussion

The ontology-based analysis of the 346 research journal publications portrays a “big picture” of the state of mental health research of adolescents and youth in Chile. To the authors’ knowledge this is the first attempt to develop such a portrait. The monad map shows that the dominant focus of the articles is on the dimension of stage followed by unit of care, condition, setting for mental health care, and population of adolescent and youth. While the research covers all the dimensions, the coverage of the elements under each dimension is not systematic. It neglects crucial elements that could play a key role in providing mental health care for adolescents and youth. The emphases and gaps in each of the dimensions are described below.

### 4.1. Stage Emphases and Gaps

Most of the research focused on identifying and assessing specific mental health conditions. The researchers found few studies that covered the stages of sensitizing, diagnosing, preventing, counseling, treating, and rehabilitating for addressing the mental health care of adolescents and youth in the Chilean context. There is a lack of intervention studies in low- and middle-income countries targeting the youth population [1]. The present analysis reiterates that similar gaps persist in Chile.

The National Mental Health Plan 2017–2025 [18] establishes among its strategic objectives: (a) to promote mental health, (b) to prevent the onset and promote early detection of mental health disorders in people and minimize the negative effects of the disease on the person, their family, and community, and (c) to increase education, providing quality information to the population on matters related to mental health, to create awareness of the importance of mental health and how to address it.

In Chile, there are effective treatments that could reduce mental health disorders, so early detection, prevention, and promotion become an ethical duty [31]. Thus, to meet these strategic objectives, it is necessary to conduct cost-effective interventions that are feasible to implement, as well as provide evidence to promote and sensitize the population about mental health strategies.

### 4.2. Unit Emphases and Gaps

The research corpus lays significant emphasis on different age groups of adolescents and youth. The literature articulates the need and relevance of mental health care for individuals across the groups of early adolescence, middle adolescence, late adolescence, and young adulthood. Given the high prevalence of mental health conditions amongst adolescents and youth, there is a greater need for formulating mechanisms for early detection [32] and the development of effective and relevant intervention programs [1].

### 4.3. Condition Emphases and Gaps

The analysis reveals that there is a greater emphasis on a certain group of mental health conditions, such as depressive disorders, anxiety disorders, substance-related disorders, trauma and stress-related disorders, impulse-control and conduct disorders, and other outcomes associated with mental health such as suicide risk, and wellbeing. However, other mental health conditions have received lesser attention, such as psychopathology of neurodevelopmental disorders, schizophrenia spectrum, bipolar disorder, obsessive-compulsive disorder, eating disorders, personality disorders, and other related effects on quality of life and stigma in mental health. These comprise the “blind spots” of the research corpus. In part, the difference in emphases may be justified by the fact that most of the studied disorders or “bright spots” have a high prevalence in the adolescence and youth population [11,33] and others, such as suicide risk, are associated with a high burden of the disease [9,10].

There is a greater need for effective research to inform practice and formulate policies in mental health conditions such as bipolar disorder, schizophrenia, and attention deficit/hyperactivity disorder (ADHD) which also have a high burden of disease in this age group [10]. These mental health conditions were also identified as a priority in the second health objective of the decade (2011–2020) in Chile [34].

### 4.4. Setting Emphases and Gaps

The analysis reflects that research on mental health care of adolescents and youth in Chile has been majorly conducted in institutional settings such as schools and universities. One easy explanation, for such a pattern, is the feasibility to recruit research subjects in these settings. There is some mention of clinics and the use of digital media which still requires further development. Digital interventions with this population have shown adequate levels of acceptability, contributing to the reduction of the gap in mental health care in developing countries [35]. There is a need to consider settings such as hospitals or clinics and reorient/specify the role of different providers such as psychiatric nurses, family, friends, volunteers, or self in providing mental health care for adolescents and youth. Disregarding the role of family and its impact on the mental health of diverse populations is a major gap in research. For instance, it has been found that underlying psychiatric disorders and family dysfunction were the main risk factors associated with suicide attempts by children [36].

### 4.5. Population Emphases and Gaps

There has been greater research emphasis on adolescents and youth in the urban population of Chile. This significantly excludes other socially disadvantaged groups such as rural, underprivileged, indigenous, disabled, LGBT, and immigrants. These groups are not a minor part of the Chilean population: national statistics suggest that 5.8% of children and adolescents (2 to 17 years) live with a disability [37], 9.5% of youth declare a sexual orientation other than heterosexual, 2.1% are trans and gender non-conforming [38] and 8.5% of youth live under poverty situation [39]. This is also reinforced by the National Mental Health Plan 2017–2025, which contemplates among its strategic objectives to protect the rights of people with mental illness who belong to vulnerable populations, in terms of access to health and social inclusion [18]. Although studying vulnerable populations could involve several challenges, its research could contribute to strengthening public policies in mental health that benefits the whole population, not just a subgroup, promoting access to health care services, social inclusion, and general wellbeing.

### 4.6. Themes Emphases and Gaps

The present state-of-the-research is selective—it does not cover all the elements of the problem, siloed—it emphasizes a few dimensions of the problem, and segmented—it focuses on partial strategies. It is not comprehensive, synoptic, and integrated. The primary theme highlights the importance of assessing urban adolescents and youth populations by institution school/university. It neglects the different stages and psychopathological conditions, ignores the concerns of rural, underprivileged, indigenous, disabled, LGBT, and immigrant populations, and the different settings that could provide mental health care services. The research corpus has widely focused on assessment of adolescents and youth mental health especially in school and university contexts. However, it has largely ignored different mechanisms of mental health care, fallen short to make systemic connections between different cohorts of population and direct research towards some of the psychopathological conditions that could affect the population, and the research has undervalued the role of different settings.

The secondary theme highlights the importance of identifying psychopathology of depressive, anxiety, and substance-related disorders. Like the primary theme, the secondary theme lacks coherence. For instance, it has been found that, depression is significantly associated with abnormal eating behavior, harmful alcohol consumption and self-destructive behavior [40]. The theme highlights the role of identification of a few psychopathological conditions and fails to make systemic connections between different levels of conditions (which can appear in terms of comorbid mental health issues) and subsequent treatment mechanisms. It further fails to represent the importance of sensitizing, diagnosing, preventing, counseling, treating and rehabilitating components of mental health care. Lastly, it underplays the inclusion of population of care and setting at which care could be provided.

The tertiary theme emphasizes only on two of the psychopathological conditions—trauma and stress-related, impulse control and conduct disorders; and two of the other conditions like suicide risk and wellbeing. For instance, there is a greater need for research to address maternal depression and associated risk behaviors in Latin America to reduce intergenerational transmission and development of adolescent depression and smoking [41]. The theme largely ignores the stage of care, different psychopathological conditions, different populations and its units, and settings for mental health care.

The quaternary theme highlights significant systematic oversights in all dimensions. Further the quaternary theme highlights the exclusion of sensitizing, diagnosing, preventing, counseling, treating, and rehabilitating- which are all critical stage functions. It fails to recognize the mental health issues and concerns of rural, underprivileged, indigenous, disabled, LGBT, and immigrants’ population. The theme also significantly exposes how research in Chilean context is skewed towards a few of the psychopathological conditions. It lastly communicates that the research has ignored the role of institutions, professionals, and others as potential stakeholders in providing mental health care and services.

In summary, the primary themes only discuss about assessing the adolescents and youth by institution school/university. The stage of care, consideration of different conditions and setting for providing care appears in the quaternary theme. The primary focus of research articles has been on identification and assessment of common mental disorders (depression and anxiety disorders) and substance-related disorders of urban adolescents and youth in the Chilean context. The role of school/university as an institution for identification and assessment purposes has been largely articulated in the studies. Directing research to explore and understand such systemic connectors or underlying factors are important for formulating comprehensive mental health care systems.

### 4.7. Limitations

This ontological analysis is insightful, but it also has limitations that are described below. While we have done our best to include the most important, current, and curated databases to find relevant articles for mapping onto the ontology, some articles might have been excluded because of non-inclusion of databases such as MEDLINE, PsycINFO, LILACS, and Latindex (the last two group Latin-American journals), and gray literature was excluded such as master’s or doctoral theses. However, new articles can always be added to the mapping incrementally and the results modified correspondingly.

Another limitation is that we do not assess the quality of the published articles, but the indexing databases considered in this study generally include journals that follow a peer review process.

### 4.8. Practical Implications and Future Directions

These results provide a synoptic view that should be presented to decision-makers to align priority-setting, and funding research on relevant mental health areas, and less explored but highly relevant and impactful areas.

The roadmap for research must be guided by the systemic framework of the ontology and systematic exploration of the strategies connoted by the pathways. Continuation of the present trajectory of research is unlikely to address the problem in Chile effectively. The trajectory must be changed. The paper provides a framework and a method to develop an effective roadmap for research to manage the mental health of adolescents and youth in Chile. Our results can also provide a roadmap for future research and policy making. For example, our findings indicate that it is important to advance research beyond identifying and assessing mental health conditions. In this regard, technology-based interventions may be a cost-effective alternative that has been little explored for prevention and early treatment in mental health in adolescents and youth in Chile [42,43]. Another example is that more research is needed on how to address stigma in mental health, given its influence as a barrier to help-seeking [44]. Interventions at the community level could also be investigated with greater emphasis, especially in minority and disadvantaged populations. As an example, the importance of spirituality in mental health outcomes has been reported in some studies in western countries, this could also be explored in Chile [45].

The method can be extended to study specific mental health conditions or populations, and to compare the status of mental health research of adolescents and youth in other countries.

## 5. Conclusions

In conclusion, this ontology shows isolated research efforts that have been carried out in Chile to explain the whole state of mental health in adolescents and youth. There is a lack of coordination between the priorities established by the decision-makers and the researchers. One of the objectives of the National Mental Health Plan 2017–2025 is to strengthen collaboration between research and policy makers for the design, implementation, and evaluation of public policies in mental health (Ministry of Health, 2017). We must generate a permanent participation and informed contribution to transform our research into knowledge that helps the discussion and development of public policies in mental health. Our results coincide with the need to strengthen mental health research in the country, prioritizing those topics that contribute to decision-making based on the needs of the population (Ministry of Health, 2017).

## Figures and Tables

**Figure 1 ijerph-19-09889-f001:**
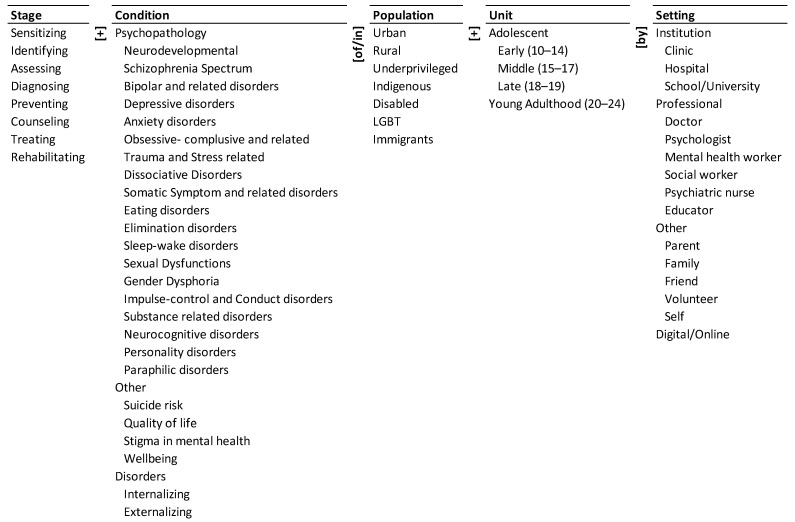
Ontology of mental health of adolescents and youth.

**Figure 2 ijerph-19-09889-f002:**
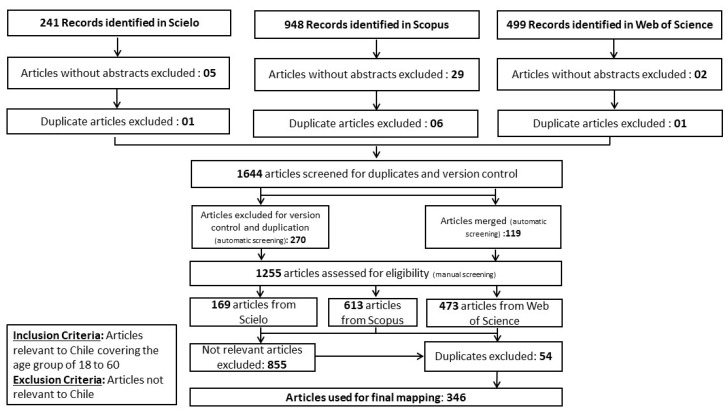
PRISMA flow diagram.

**Figure 3 ijerph-19-09889-f003:**
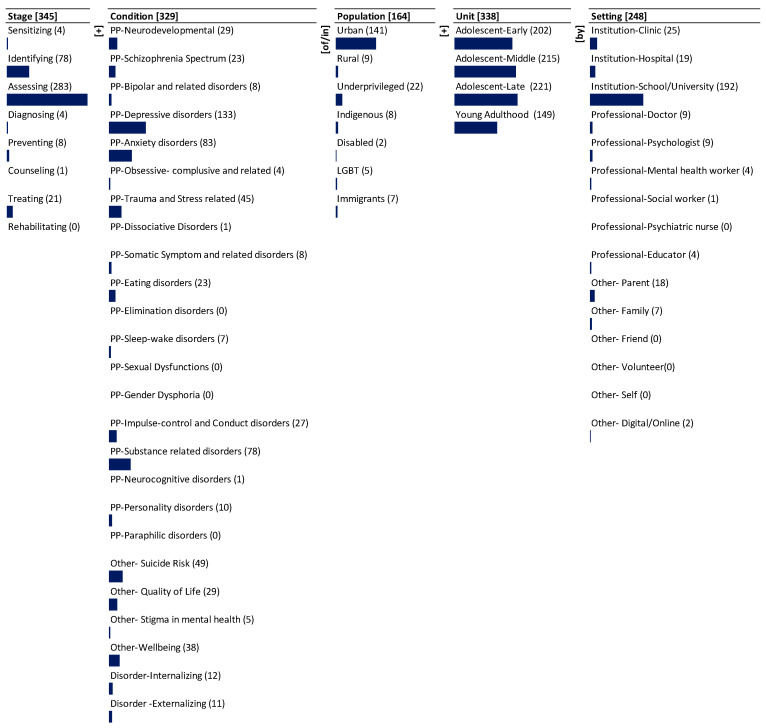
Monad map of mental health of adolescents and youth in Chile.

**Figure 4 ijerph-19-09889-f004:**
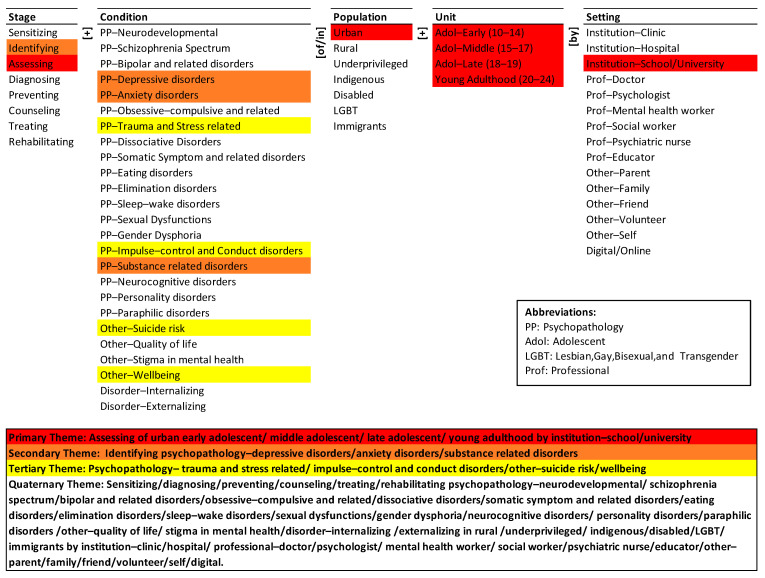
Theme map of mental health of adolescents and youth in Chile.

## Data Availability

The datasets used and/or analyzed during the current study are available from the corresponding author upon reasonable request.

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
