# Peer review of "State of Mental Health Research of Adolescents and Youth in Chile: An Ontological Analysis"

_ijerph, 2022, doi:10.3390/ijerph19169889_

Round 1

Reviewer 1 Report

The reviewed article is a literature review and focuses on the mental health of the adolescent and youth population of Chile. This is a very good paper that gives the opportunity not only to describe mental health in Chile but globally by presenting an ontological framework, thus I recommend it for publication. It is well organized and describes the main health problems of adolescents and youths. 

Suggested corrections are listed below:

1. the First comment is regard to Keywords as it lacks the information that only research on the population from Chile was analyzed. Hence it gives the impression that the adolescent and youth mental health studies conducted all over the world are analyzed.

2. In the paragraph in lines 46-59 some thoughts are doubled I suggest re-editing it

3. In paragraphs lines 156-162 you doubled the information about the “Stage” and “Settings” columns – it is unnecessary

4. The authors in lines 245-262 referred to the different colors (e.g. brown, yellow) of themes in fig, 4 however only two colors are presented in this figure red and green so this must be corrected

5. The interpretations of the representations in each theme are rather residual and the expression “it is simple” or “it too is simple”  is confusing – I suggest describing it in more scientific language, more precisely. (lines245-261)

6. Discussion section – overall the more in-depth interpretations and practical conclusions with examples from different countries as ways of implementing good practices and policy for each gap/point referred to in the discussion are recommended. At the moment only general information is given and most conclusions that bring nothing specific to the discussion are given.

7. In the discussion section the last paragraph of subsection 4.1 the last sentence is: “Thus, to meet this  strategic objective, it is necessary to conduct cost-effective interventions that are feasible  to implement in our context, as well as provide enough evidence to promote and sensitize  the population about mental health strategies.” The expressions “in our context” or “provide enough evidence” are not clear, I also advise to describe in detail these conclusions

6. Small editorial errors should be corrected e.g. doubled spaces throughout the entire paper (lines 93, 125, 170, 177, 201, 203, etc.);  capital letter instead of lower case (line 117 word “Identifying”

7. Linguistic proofreading is recommended because authors often do not follow the sequence of time or use stylistically incorrect constructions

Author Response

Dear reviewer

Thanks for your comments.

Reviewer 2 Report

The content of the article is messy and the discussion section lacks focus.

Author Response

Dear reviewer

Thanks for your comment.

The entire text was revised to make it more understandable and the discussion was improved by adding concrete examples.

Reviewer 3 Report

I would like to thank the Editor for the opportunity to review this study and I am flattered to be able to provide my contribution. In general, I find this article to be well written, I do find this paper to be a good discussion issue about mental health problems in Chile. However, the paper presents some weaknesses, and I would suggest reconsidering publishing this manuscript after minor revisions. Suggestions are reported in the following comments. I would ask the Authors to address minor amendments, as follows:

1. Introduction. It would be desirable for the authors to include literature on mental problems from Chile and/or other Spanish-speaking countries. The theoretical framework is very much focused on literature from English-speaking countries.

2. Method. They should justify the reasons why they have not included medline among the databases. In any case, they should include its absence as a limitation of the work.

3.        Results. On page 7, line 223, there is an error. Studies about late adolescence (215). In Figure 3, say "late adolescence (221). Correct this.

I hope these comments are useful in moving your research forward.

Author Response

Dear reviewer 

Thanks for your comments.

  1. Introduction. It would be desirable for the authors to include literature on mental problems from Chile and/or other Spanish-speaking countries. The theoretical framework is very much focused on literature from English-speaking countries Response: A paragraph on the Chilean context was added.
  2. Method. They should justify the reasons why they have not included medline among the databases. In any case, they should include its absence as a limitation of the work. Response: We add it as a limitation.
  3. Results. On page 7, line 223, there is an error. Studies about late adolescence (215). In Figure 3, say "late adolescence (221). Correct this. Response: We have corrected the error.

Reviewer 4 Report

Dear Authors - your study has set out what it intended to do. My comments are to direct mental health problems to include the field of human consciousness and spirituality - you can mention this as an aside if you feel it appropriate. 

Author Response

Dear reviewer

We are grateful for the reflections made on the subject. We have incorporated the topic of spirituality into the discussion.

We incorporate wellbeing and quality of life as part of the broad definition of mental health beyond psychopathology. And because low wellbeing and quality of life is related to poorer mental health.

Round 2

Reviewer 2 Report

I think this article is very interesting, as for the design of the article, the application of statistics, I think I have nothing to criticize. This article would be more perfect if the references were newer and more authoritative. Again, the discussion should be a little more concise, a little too long

Author Response

Dear Reviewer
Thank you for your comments.
Regarding the references, more than 60% of them are from the last 5 years. The other references although older are relevant and widely cited in the literature. One reviewer asked us to add references from the Chilean context, which we did.
In relation to the discussion, it was edited again and shortened, but it should be considered that another reviewer had asked us to add examples, which was also done.